# Spatial Econometric Analysis of the Impact of Socioeconomic Factors on PM_2.5_ Concentration in China’s Inland Cities: A Case Study from Chengdu Plain Economic Zone

**DOI:** 10.3390/ijerph17010074

**Published:** 2019-12-20

**Authors:** Ye Yang, Haifeng Lan, Jing Li

**Affiliations:** 1School of Architecture and Urban-Rural Planning, Sichuan Agriculture University, Dujiangyan 611830, China; yangye@stu.sicau.edu.cn; 2School of Civil Engineering and Architecture, Southwest University of Science and Technology, Mianyang 621010, China; lanhaifeng@mails.swust.edu.cn

**Keywords:** spatiotemporal distribution, socioeconomic factors, spatial econometrics, PM_2.5_ concentration, spillover effects

## Abstract

Particulate matter with a diameter less than 2.5 µm (PM_2.5_), one of the main sources of air pollution, has increasingly become a concern of the people and governments in China. Examining the socioeconomic factors influencing on PM_2.5_ concentration is important for regional prevention and control. Previous studies mainly concentrated on the economically developed eastern coastal cities, but few studies focused on inland cities. This study selected Chengdu Plain Economic Zone (CPEZ), an inland region with heavy smog, and used spatial econometrics methods to identify the spatiotemporal distribution characteristics of PM_2.5_ concentration and the socioeconomic factors underlying it from 2006 to 2016. Moran’s index indicates that PM_2.5_ concentration in CPEZ does have spatial aggregation characteristics. In general, the spatial clustering from the fluctuation state to the stable low state decreased by 1% annually on average, from 0.190 (*p* < 0.05) in 2006 to 0.083 (*p* < 0.1) in 2016. According to the results of the spatial Durbin model (SDM), socioeconomic factors including population density, energy consumption per unit of output, gross domestic product (GDP), and per capita GDP have a positive effect on PM_2.5_ concentration, while greening rate and per capita park space have a negative effect. Additionally, those factors have identified spatial spillover effects on PM_2.5_ concentration. This study could be a reference and support for the formulation of more efficient air pollution control policies in inland cities.

## 1. Introduction

In recent years, China’s particulate matter with a diameter less than 2.5 µm (PM_2.5_) pollution has attracted public attention. PM_2.5_ pollution can not only make the urban atmosphere hazy, but also cause overall health damage to humans [1,2]. Although public discussions about PM_2.5_ pushed it into the new national standard and governments around China have made varying degrees of progress in disclosing PM_2.5_ information, there is no clear agreement on what are the main sources and how to take care of PM_2.5_ pollution.

In China, some existing studies have indicated that natural conditions, such as temperature [3], precipitation [4], wind speed [5,6], wind direction [7], terrain [8] are important factors affecting the accumulation and diffusion of PM_2.5_. Besides natural conditions, a growing number of scholars have explored the correlations between PM_2.5_ concentration and socioeconomic factors, including population [9,10], industrial structure [11], per capita GDP [12], energy consumption [13,14], vehicle population [15], and so on, demonstrating that human activities are the fundamental causes of high PM_2.5_ concentration. 

For instance, Yan [9] used a spatial interpolation method and spatial clustering analysis to explore the evolution of spatiotemporal patterns of PM_2.5_ concentration in the Beijing–Tianjin–Hebei (BTH) region and found that the region has significant spatial autocorrelation due to high population density. Moran’s index (Moran’s I) analysis and regression analysis were used by Zhao [15] to detect the spatial autocorrelation of PM_2.5_ pollution in 289 cities in China and the relationship between PM_2.5_ pollution and the variables. That study concluded that vehicle population has the most significant effect on PM_2.5_ concentration. Yun [10] adopted the spatial statistical analysis and the establishment of a geographic detector model to explore the concentrations of PM_2.5_ in Yangtze River Delta (YRD) from 2005 to 2015, which revealed that population density is the key factor affecting the concentrations of PM_2.5_. Xiao [11] used the CAMx (v5.4) modeling system to explore the nexus between the variables and PM_2.5_ emissions in the Pearl River Delta (PRD), and the research showed that car ownership, average travel distance, and industrial production are the major contributors to PM_2.5_ in PRD. Yang’s [12] study indicated that PM_2.5_ concentration is significantly positively spatially correlated with GDP per capita, industrial added value, private car ownership, and urban population density. More details about those studies are summarized in Table 1.

However, those studies were mainly concentrated in BTH [9], YRD [10], and PRD [11], which are eastern coastal areas of China with high population density and developed economy, while few studies have been conducted in inland cities of China. There are great differences between coastal cities and inland cities in terms of geographic conditions [16], climatic conditions [17], socioeconomic background [18,19]. As a result, it may be less useful and even unreasonable to take lessons from the research results of coastal cities on the spatiotemporal variation of PM_2.5_ to make air pollution control policies for inland cities. Consequently, as a typical inland urban agglomeration of China and an area regarded as one of the four major smog regions [20], CPEZ was selected as the research area. In this paper, the following main questions need to be answered:(1)What are the spatiotemporal distribution characteristics, regional differences, and variation trends of PM_2.5_ in CPEZ?(2)What is the influence of socioeconomic factors on PM_2.5_ concentration in CPEZ and how does it work?(3)What are the policy implications for the formulation of PM_2.5_ pollution control in CPEZ as well as other inland cities?

## 2. Materials and Methods

### 2.1. Study Area and Data Sources

As one of strongest and largest economic and population intensive areas in western China [21], CPZE covers eight cities—Chengdu, Deyang, Mianyang, Meishan, Leshan, Ziyang, Suining, and Ya’an—with a total area of 81,300 square kilometers and a population of more than 40 million. It is located in the eastern margin of the Western Sichuan Plateau and the Sichuan Basin (Figure 1). Stable weather is easily formed by this geographic environment, which means it is not conducive to the diffusion and dilution of pollutants and aggravates air pollution. Specifically, because CPEZ is located in the basin topography, the atmospheric environment capacity is very limited and where prolonged breezes or calm winds in the area can inhibit advection transport of pollutants and hinder their diffusion, which would multiply the amount of pollutants near the ground [22] Besides, the phenomenon of temperature inversion in urban is serious, that is the upper air temperature is higher than the lower air temperature. Once the formation of this inversion, the air cannot convection up and down, which is difficult to diffuse pollutants [22,23]. Those are why it is one of the 4 regions with the worst smog in China [21]. In addition, the economic development and population growth caused by the influx of a large number of people led to huge problems in the air environment in CPEZ. 

In this study, data on the average yearly PM_2.5_ concentration in the 8 cities from 2006 to 2016 were collected from the Socioeconomic Data and Applications Center (SEDAC) of Columbia University (https://beta.sedac.ciesin.columbia.edu/data/set/sdei-global-annual-avg-pm2-5d). As shown in Figure 2, there is obvious diversity among the cities. For example, Chengdu’s PM_2.5_ concentration is clearly higher than the other cities of CPEZ. In addition, socioeconomic data were derived from China city statistical annual reports and Sichuan provincial statistical annual reports from 2006–2016. All factors selected for this paper are explained in detail in the following section on the STIRPAT model [24].

### 2.2. Spatial Autocorrelation Analysis

In this paper, global Moran’s I was used to test the global spatial autocorrelation of PM_2.5_ concentration. The calculation of global Moran’s I is shown in Formula (1). At the same time, local Moran’s I was used to identify the local spatial autocorrelation of atmospheric PM_2.5_ pollution and the location of spatial agglomeration and spatial heterogeneity, as shown in Formula (2). If Moran’s I is greater than 0, this indicates that the study object has positive spatial autocorrelation of PM_2.5_, and the larger the value, the stronger the spatial clustering. If Moran’s I is less than 0, this indicates that PM_2.5_ concentration has a negative spatial autocorrelation relationship, and the smaller the value, the stronger the spatial dispersion of the observed value.
(1)$$I_{G}=\frac{\sum_{i=1}^{n}\sum_{j=1}^{n}W_{ij}\left(x_{i}-\bar{x}\right)\left(x_{j}-\bar{x}\right)}{S^{2}\sum_{i=1}^{n}\sum_{j=1}^{n}W_{y}}$$(2)$$I_{L}=\frac{\sum_{i=1}^{n}\sum_{j=1}^{n}W_{ij}\left(x_{i}-\bar{x}\right)\left(x_{j}-\bar{x}\right)}{S^{2}}$$
where *I_G_* is global Moran’s I, *I_L_* is local Moran’s I, n is the number of spatial units, *x_i_* and *x_j_* are the PM_2.5_ annual average concentration values of units *i* and *j*, respectively, and $\bar{x}$ is the average value of all units. *S* is the standard deviation. *W_ij_* is the spatial weight matrix of elements *i* and *j*. If there is a common edge between spatial elements *i* and *j*, *W_ij_* = 1, otherwise *W_ij_* = 0.

The significance level of local Moran’s I can be measured by Z (I), and its calculation formula is shown as Formula (3). By comparing the sign of Z (I) and the significance level of Moran’s I, the spatial units can be divided into 4 types of spatial autocorrelation relations. First, if Moran’s I is significantly positive and Z (I) > 0, it is a “high–high” type, that is, the PM_2.5_ concentration values of this unit and adjacent units are relatively high. Second, if Moran’s I is significantly positive and Z (I) < 0, it is a “low–low” type, that is, the concentration values of PM_2.5_ in this unit and its neighboring units are relatively low. Third, if Moran’s I is significantly negative and Z (I) > 0, it is “high–low” type, and units with high PM_2.5_ concentration value are surrounded by adjacent low-value units. Finally, if Moran’s I is significantly negative and Z (I) < 0, it is “low–high” type, and units with low PM_2.5_ concentration value are surrounded by adjacent high-value units:(3)$$Z\left(I\right)=\frac{\left[I-E\left(I\right)\right]}{\sqrt{Var\left(I\right)}}$$
where Z (I) measures the significance level of Moran’s I, *E (I)* is the mathematical expectation of global Moran’s I, and *Var (I)* is the variance of global Moran’s I.

### 2.3. Socioeconomic Factor Selection

The STIRPAT model is a classic theoretical framework for the study of factors that influence environmental pollution [25]. The advantage of the STIRPAT model is that it can estimate each variable’s coefficient and the impact factors can be modified [26]. To be consistent with most research, this paper adopted the STIRPAT model proposed by Dietz and Rosa (1998) as the basic theoretical framework. The standard form of the STIRPAT model is shown as Formula (4). It can be seen that environmental quality is related to population size, affluence level, and technological level. Based on this and related literature about socioeconomic factors that can affect PM_2.5_ concentration, we added urban environment factors (E), including urbanization rate, green rate, and per capita park green space, as shown in Formula (5). Table 2 shows information on all selected variables: full name, abbreviation definition, unit, types, and reference; statistical descriptions of those variables in the 8 cities of CPEZ can be seen in the Appendix A
Table A1:(4)$$Ln\left(I_{it}\right)=a+blnP_{it}+clnA_{it}+dlnT_{it}+\varepsilon_{it}$$
where *I_it_* represents environmental quality at location *i* at time t, *P* represents the size of the population at location *i* at time *t*, *A* represents affluence level at location *i* at time *t*, *T* represents the technical level at location *i* at time *t*, and *ε* is an error term:(5)$$\begin{array}{c} {\mathrm{lnPM}}_{2.5}=\propto +\beta_{1}lnPD+\beta_{2}lnGDP+\beta_{3}lnGDPP+\beta_{4}lnSIR+\beta_{5}lnEC+\beta_{6}lnBR\\ +\beta_{7}lnGR+\beta_{8}lnPP+\varepsilon_{it} \end{array}$$

### 2.4. Spatial Econometric Model

When the data involve geospatial features, the observed values cannot remain independent because closer distance may cause relevance [38]. If the spatial effects are neglected in the econometric model, the estimation results will be biased [33]. Consequently, spatial weight is introduced to adjust the relationships between independent variables, dependent variables, and residual terms and dependent variables to reflect spatial interaction relations. For example, common spatial interaction relations include endogenous interactions between dependent variables (Spatial lag model (SLM)), interactions between error terms (spatial error model (SEM)), and based on SLM adding exogenous interactions between independent variables (spatial Durbin model (SDM)), as shown in Figure 3. 

Compared with other spatial econometric models, the spatial Durbin model can explain not only the influence of variables on the unit itself, but also the influence of other variables of adjacent units on that unit. It considers a more comprehensive impact and has stronger explanatory power [39]. Therefore, this paper adopts the spatial Durbin model to answer the research questions. The formula is shown as Equation (6):(6)$$y_{it}=a+\rho \overset{n}{\underset{j=1}{\sum }}w_{ij}y_{it}+x_{it}\beta +\overset{n}{\underset{j=1}{\sum }}w_{ij}x_{it}\theta +u_{i}+v_{t}+\varepsilon_{it}$$
where *i* is location, *t* is time, *ρ* is the spatial autocorrelation coefficient of the dependent variable, *β* and *θ* are the correlation coefficients of the independent variable, *w_ij_* is a spatial weight matrix, *u_i_* is the spatial residual error, *v_i_* is the time residual error, and *ε_it_* is the residual error of time and space.

Furthermore, to make the model easy to understand, Lesage and Pace [40] divided the effects in SDM into direct effects (DEs), indirect effects (IEs), and total effects (TEs) (Equations (7)–(12)). DE refers to the influence of independent variables on the dependent variable in the region. IE, also known as the spatial spillover effect, is used to measure the impact of an independent variable in an adjacent region on the dependent variable in the region.
(7)$$y_{t}={\left(I-\rho w\right)}^{-1}\left(\alpha +x_{t}\beta +W_{x_{t}}\theta +\lambda_{t}\iota_{n}+\varepsilon_{t}\right)$$(8)$$E\left(y_{t}|x_{t}\right)={\left(I-\rho W\right)}^{-1}\left(\alpha +x_{t}\beta +Wx_{t}\theta +\lambda_{t}\iota_{n}\right)$$(9)$$\left[\begin{array}{ccc} \frac{\partial E\left(y_{1}\right)}{\partial x_{1}} & \cdots & \frac{\partial E\left(y_{1}\right)}{\partial x_{n}}\\ \vdots & \ddots & \vdots \\ \frac{\partial E\left(y_{n}\right)}{\partial x_{1}} & \cdots & \frac{\partial E\left(y_{n}\right)}{\partial x_{n}} \end{array}\right]={\left(I-\rho W\right)}^{-1}\left[\begin{array}{cccc} \beta_{k} & \omega_{12}\theta_{k} & \cdots & \omega_{1n}\theta_{k}\\ \omega_{21}\theta_{k} & \beta_{k} & \cdots & \omega_{2n}\theta_{k}\\ \vdots & \cdots & \ddots & \vdots \\ \omega_{n1}\theta_{k} & \omega_{n2}\theta_{k} & \cdots & \beta_{k} \end{array}\right]$$(10)$$TE=\left\{{\left(I-\rho W\right)}^{-1}\times \left(\beta_{k}I+\theta_{k}W\right)\right\}$$

Direct effect (DE) is the average value of the diagonal elements of the above matrix. If $A^{\bar{d}}$ represents the row average of the diagonal elements of matrix A, DE can be expressed as:(11)$$DE=\left\{{\left(I-\rho W\right)}^{-1}\times {\left(\beta_{k}I+\theta_{k}W\right)}^{\bar{d}}\right\}$$

The indirect effect is the row average of nondiagonal matrix elements. If $A^{\bar{rsum}}$ represents the row average of matrix A’s nondiagonal elements, IE can be expressed as:(12)$$IE={\left\{{\left(I-\rho W\right)}^{-1}\times \left(\beta_{K}I+\theta_{K}W\right)\right\}}^{\bar{rsum}}$$

## 3. Results

### 3.1. Spatiotemporal Variation of PM_2.5_

The changing pattern of PM_2.5_ concentration distribution in CPEZ from 2006 to 2016 can be clearly seen in Figure 4. Although the value of PM_2.5_ in each city changed during the period, these cities can be generally divided into three groups according to the PM_2.5_ values, as high, medium, and low. Specifically, the first group was represented by Chengdu, with a PM_2.5_ concentration of more than 50 µg/m^3^, which was higher than in other cities. The second group was located near eastern Chengdu (Deyang, Suining, Ziyang, and Meishan), where the PM_2.5_ was about 30–50 ug/m^3^. The third group, with a PM_2.5_ concentration lower than 30 µg/m^3^, was Mianyang, Leshan, and Ya’an. In general, the closer to Chengdu, the higher the PM_2.5_ concentration, which means that PM_2.5_ concentration had the characteristic of spatial agglomeration. According to air quality guidelines issued by the World Health Organization (WHO) in 2005, the average daily concentration should not exceed 25ug/m^3^, otherwise it should be considered unsafe living conditions. It seems that most cities in CPEZ were not up to the standard of health, but the PM_2.5_ pollution tended to be moderate in recent years. According to the PM_2.5_ concentration from 2006–2016 in CPEZ, global Moran’s I was calculated, shown in Table 3. In 2006, Moran’s I of PM_2.5_ in CPEZ was 0.190 (*p* < 0.05), indicating that PM_2.5_ concentration showed spatial aggregation. However, with the passage of time, such spatial aggregation characteristics gradually weaken year by year (average 1% reduction per year). At the end of 2016, Moran’s I of PM_2.5_ in CPEZ was 0.083 (*p* < 0.1), which was less than half of 2006. 

In order to further explore the changes of spatial aggregation of PM_2.5_ concentration with time, a cluster and outlier analysis of PM_2.5_ concentrations from 2006-2016 is adopted, as shown in Figure 5. By the figure we can clearly see that the cluster areas with statistical significance (*p* < 0.05) are mainly Chengdu and cities in the east of Chengdu such as Deyang, Ziyang and suining. While, the outlier areas with statistical significance (*p* < 0.05) are mainly located at the edge of CPEZ, where is relatively far from Chengdu. From the perspective of temporal variation trend, the spatial distribution of cluster and outlier before 2012 showed an obvious change state. Almost every year is different. While in the recent four years, the spatial distribution of cluster and outlier showed a stable state.

The Moran scatter plot [41] is a useful visual tool for exploratory analysis, because it enables you to assess how similar an observed value is to its neighboring observations. Its horizontal axis is based on the values of the observations and is also known as the response axis. The vertical Y axis is based on the weighted average or spatial lag of the corresponding observation on the horizontal X axis.

Through the scatter plot of Moran’s I based on local Moran’s I analysis (Figure 6), it can be seen that most of the scatter is located in the first quadrant (Chengdu, Deyang, Suining, and Ziyang). 

This indicates that areas with high PM_2.5_ tend to be adjacent to high PM_2.5_ areas. In the third quadrant (Leshan), low PM_2.5_ areas tend to be adjacent to low PM_2.5_ areas. The clusters of “high–high” and “low–low” reflect a positive spatial correlation of PM_2.5_ in the different cities of CPEZ. However, there are still a few clusters in the second and fourth quadrants (Meishan, Mianyang, and Ya’an), meaning the low-level area is encircled by the surrounding high-level area. From the perspective of the entire period from 2006 to 2016 (Figure 7), local Moran’s I of Leshan, Ziyang, and Deyang is greater than 0, which means that PM_2.5_ tends to form spatial aggregation and would lead to much worse air pollution. In Chengdu, Meishan, and Suining, local Moran’s I is approximately equal to 0, which means the PM_2.5_ of these areas is relatively stable, neither aggregating nor diffusing. However, local Moran’s I in the Ya’an and Mianyang areas is less than 0, and the PM_2.5_ concentration in these regions is relatively low, trending toward scattered spatial aggregation.

### 3.2. Spatial Econometric Regression

The purpose of the model (SDM) is to identify the impact of social-economic variables on PM_2.5_ concentration which contain the dimensions of space and time, but we cannot determine directly whether the difference in PM_2.5_ concentration caused by time and space is random (random effect) or presenting a certain regularity (fixed effect), so we adopt four models for comparison and a brief introduction of those models as follows:(1)SDM time fixed effect: for different spatial individuals, differences caused by time are consistent.(2)SDM spatial fixed effect: among cross-sectional data of different time series, differences caused by spatial characteristics are consistent.(3)SDM time and spatial fixed effect: among cross-sectional data of different time series, differences caused by space are consistent, and among different spatial individuals, differences caused by time are consistent.(4)SDM random effect: the differences caused by space and time is random.

When comparing R2 and AdjR2 of the regression results of the four effects in Table 4, the fitting degree of model 1 (SDM time fixed effect) is higher than that of other models. Therefore, this paper uses this model to identify the spatiotemporal distribution characteristics of PM_2.5_ concentration. The direct effects of regression results demonstrate that population density, energy consumption per unit of output, and per capita park area are significant socioeconomic variables that influence PM_2.5_ concentration (*p* < 0.01) in CPEZ. The relatively significant indicators (*p* < 0.05) are regional GDP and urban green rate, and the slightly significant variable (*p* < 0.1) is per capita GDP, while the proportions of secondary industry (SIR) and built-up areas are insignificant variables for PM_2.5_ concentration. Among the socioeconomic variables, population density, per capita GDP, and energy consumption per unit of output have a positive effect on PM_2.5_ concentration, that is, as these variables increase, PM_2.5_ concentration will increase, whereas urban green rate and per capita park area have a negative effect on PM_2.5_ concentration, indicating that increasing the urban green rate and per capita park areas will alleviate PM_2.5_ pollution.

The effect decomposition results are shown in Table 5. In fact, except for the degree of impact, the result of indirect (spillover) effects shows similar results to the direct effect, that is, population density, per capita GDP, and output value of energy consumption have a positive effect on PM_2.5_ concentration, while greening rate and per capita park area have a negative effect. However, it has a discriminative implication, which indicates that apart from a local city’s socioeconomic variables, the socioeconomic variables of neighboring cities also effect the PM_2.5_ concentration of the local city. What is meant by this is that in CPEZ, the socioeconomic influence factors of both a particular city and its neighboring cities drive the city’s change of PM_2.5_ emissions.

## 4. Discussion

The Moran’s I of PM_2.5_ concentration in CPEZ is about 0.08–0.19, which reveals an autocorrelation of PM_2.5_ concentration in the region. However, Moran’s I of PM_2.5_ concentration in CPEZ is relatively lower compared with the three major economic growth areas (BTH, YRD, and PRD), which is about 0.4 to 0.9 [42,43,44]. There are two possible reasons for this situation. For Economic reason, it can be seen from Appendix A
Table A1 that the GDP of Chengdu is far higher than that of other cities in CPEZ. Because of the positive correlation between the economy and PM_2.5_ pollution [45,46], unbalanced economic development also leads to unbalanced PM_2.5_ distribution, resulting in the weak spatial correlation of CPEZ. Compared with Chengdu, the economic development of cities in the PRD, YRD, and BTH regions is more balanced. For Geography reason, unlike the PRD, YRD, and BTH regions, CPEZ is located in a plain of the Sichuan Basin. This kind of terrain makes it difficult for the central city group to form air flow with its peripheral cities [22,23,47]. It is reflected in Moran’s I that the central city group has strong correlation, but the overall correlation of all cities is not strong.

Among the factors related to PM_2.5_ concentration, previous studies by Zhang [48], Yun [10], Xie [49], and Ding [50] also showed a significant positive correlation between population density and PM_2.5_; in addition, Guo and Ding showed that population density is the most important socioeconomic factor affecting PM_2.5_ concentration [51,52]. This study also confirms their results with similar findings. Additionally, our study found that there is a significant positive correlation between energy consumption and PM_2.5_ concentration, which is consistent with the existing results that higher energy consumption leads to higher PM_2.5_ concentration [53,54]. Furthermore, this study identified that GDP and per capita GDP have a positive effect on PM_2.5_ concentration. Some studies also claimed that economic development will lead to increased PM_2.5_ concentration [55,56,57] But Wang’s [58] study revealed a negative relationship between per capita GDP and PM_2.5_ in southeast China (the most developed part of the country) and backward areas in China. He argued that the main reason for the different results is different development levels of Chinese cities leading to varying PM_2.5_ profiles. Moreover, there is a significant negative correlation between PM_2.5_ and per capita park green space and urban greening rate, and their growth can effectively reduce PM_2.5_ concentration. From an ecological point of view, green spaces in urban areas can absorb and purify PM_2.5_, which can improve air quality [59]. The existing research also shows that there is a significant relationship between these factors and PM_2.5_ [28].

Besides that, in our study, there are two factors that are not relevant: the proportions of secondary industry and built-up areas. However, in studies of 289 cities in China [37], YRD [36], Bohai Rim Urban Agglomeration [60], and PRD [61], it was shown that the proportions of secondary industry and built-up areas are significantly related to PM_2.5_. In fact, different types of cities will face different problems in the development process [62]. Yan’s [63] research showed that the industrial results need to exceed the threshold value to have an impact on PM_2.5_, while the secondary industry in CPEZ is weaker than that in the YRD, PRD, and BTH regions, so the impact of the proportion of secondary industry on PM_2.5_ is too low to be significantly reflected.

Finally, in order to make the comparison between CPEZ and BTH [9], YRD [10], and PRD [11] more intuitive, we sorted out the regression coefficients obtained from relevant studies according to population, affluence level, technical level and urban environment, and normalized the results, as shown in Figure 8. It is clearly that although there are differences in the weight of influencing factors in each region, population factor is the most important influencing factor. Addtionally, in CPEZ the weight of affluence level factor is higher than that of other regions, while the weight of technology level factor is lower than that of other regions. Which means the economy of CPEZ is less value-added and energy efficient than elsewhere and the development of green economy and high value-added industries needs to be strengthened. The influence factors of urban environment lie in the middle level between these regions, indicating that the urban ecological construction performs well but still has room for improvement.

## 5. Conclusions

In this paper, we studied temporal and spatial patterns of PM_2.5_ concentration in CPEZ, an inland urban agglomeration of China. Global Moran’s I was used to analyze spatiotemporal variations in the region, while the STIRPAT model and spatial econometrics method were applied to explain the spatial heterogeneity of regional PM_2.5_ concentration in CPEZ over the study period from 2006 to 2016. A better understanding of the identified driving factors and their impacts on PM_2.5_ pollution may be useful for policy-makers in implementing PM_2.5_ pollution control policies.

The Moran’s I of PM_2.5_ concentration over the study period indicated that PM_2.5_ concentration in CPEZ did show spatial aggregation and was risks to human health in a majority of cities in CPEZ. In general, the closer to Chengdu, the higher the PM_2.5_ concentration. In addition, from 2006 to 2016, the spatial aggregation characteristics showed that the initial high and low fluctuation states gradually changed to stable low fluctuation states.

Among socioeconomic factors, population density, per capita GDP, and output value of energy consumption have a positive effect on PM_2.5_ concentration, while greening rate and per capita park area have a negative effect. We also recognized the significance of the spatial spillover effect in regional air pollution control.

Based on the findings above, we propose some policy recommendations as follows. Due to the existing spatial autocorrelations and spatial spillover effect between regions, the government should pay attention to the importance of regional joint governance mechanisms in the PM_2.5_ governance process. This means that implementing environmental regulations in a separate region cannot bring sufficient benefits to a region without emphasizing regional linkages of environmental regulations. Additionally, it is necessary to promote the use of clean energy, increase the added value of energy consumption, and realize the green transformation of the economy. Moreover, different from the treatment methods of reducing pollution sources and controlling PM_2.5_ emissions, measures such as low-carbon cities, forest cites, and ecological cities, which take advantage of the biological characteristics of plants to absorb and retain particles in the atmosphere, would also be of benefit for reducing and controlling the particle content in the atmosphere.

Although the results fill a research gap in the inland cities and put forward a range of meaningful suggestions, there are still some deficiencies that mainly in two aspects. On the one hand, the selection of factors only emphasizes the role of socio-economic impact factors, while ignoring the role of physical environment variables such as wind, temperature and terrain. On the other hand, the number of cities or scope of study area is small, so more samples are needed for further research in the future to improve the credibility of the research results.

## Figures and Tables

**Figure 1 ijerph-17-00074-f001:**
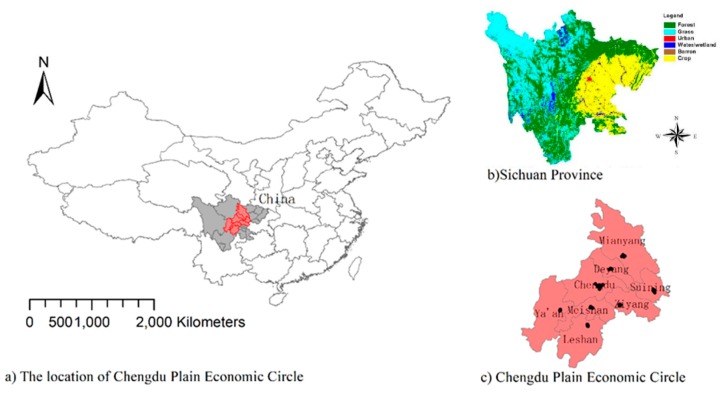
The location of CPEZ

**Figure 2 ijerph-17-00074-f002:**
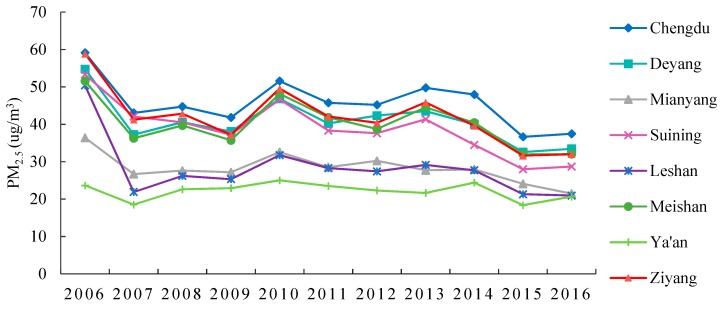
The variation of PM_2.5_ concentration over 2006 to 2016.

**Figure 3 ijerph-17-00074-f003:**
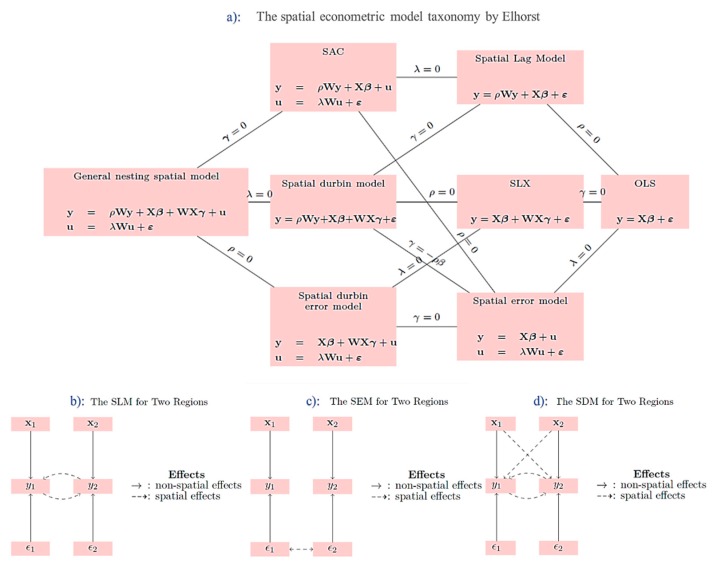
Spatial economic model taxonomy

**Figure 4 ijerph-17-00074-f004:**
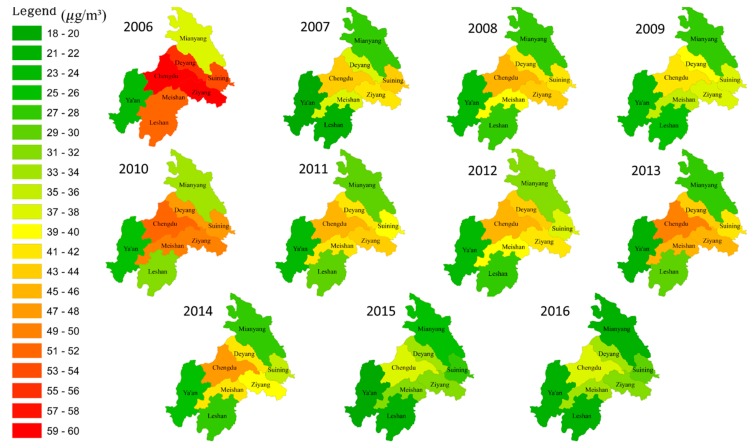
Changing pattern of PM_2.5_ distribution in the CPEZ from 2006 to 2016.

**Figure 5 ijerph-17-00074-f005:**
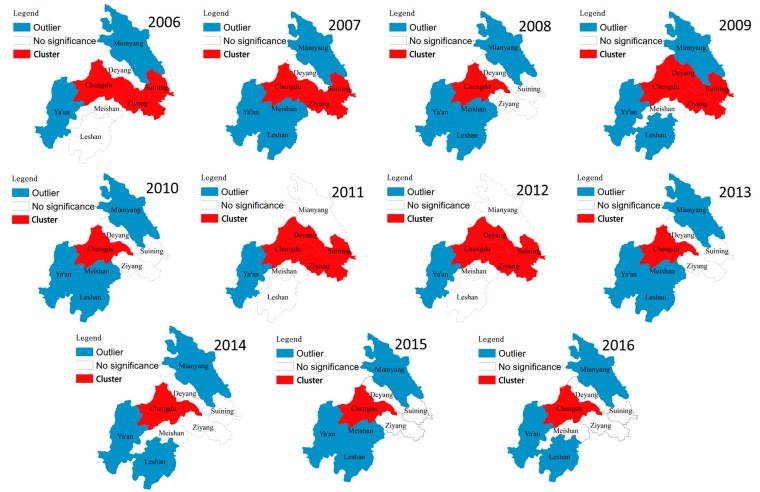
Cluster and outlier analysis of PM_2.5_ concentration in the CPEZ from 2006 to 2016.

**Figure 6 ijerph-17-00074-f006:**
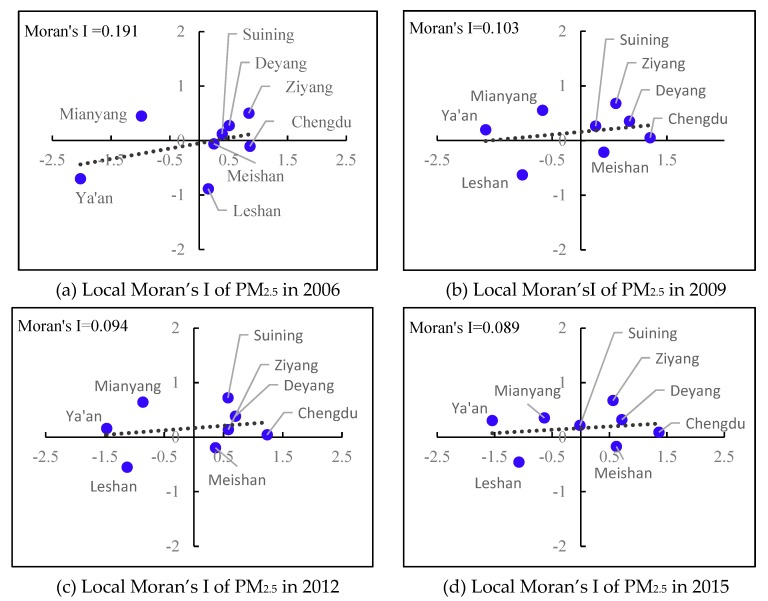
Local Moran’s I of eight cities in CPZE.

**Figure 7 ijerph-17-00074-f007:**
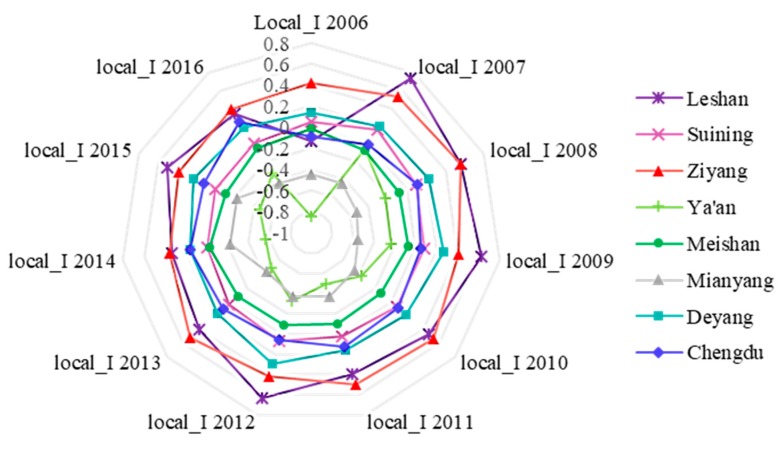
Local Moran’s I variation from 2006 to 2016

**Figure 8 ijerph-17-00074-f008:**
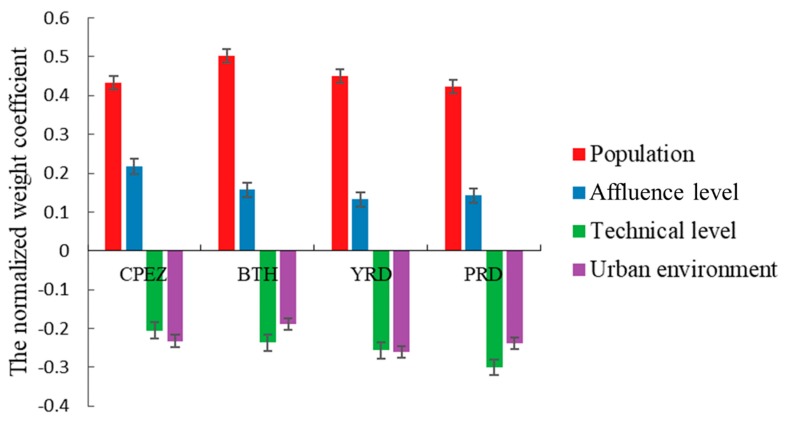
The weight comparison of influencing factors in different regions

**Table 1 ijerph-17-00074-t001:** A summary of relevant studies.

References	Time	Location	Socioeconomic Variables	Methodologies	Key Findings
Dan Yan [9]	2018	BTH	Population density, Energy structure, urbanization	Spatial interpolation method, spatial clustering analysis.	PM_2.5_ in BTH region has significant spatial autocorrelation due to high population density.
Shen Zhao [15]	2019	289 Chinese cities	Human activity intensity, the secondary industry’s proportion, emissions of motor vehicles.	Spatial clustering analysis, regression analysis.	vehicle population is the most critical driver of increasing PM_2.5_ concentration
Guoliang Yun [10]	2019	YRD	Population density, GDP	Geographical detector model.	Population density is the dominant socioeconomic factors affecting the formation of PM_2.5_.
Xiaohong Yin [11]	2016	PRD	Vehicle ownership; industrial production; residential; travel distance.	CAMx (v5.4) modeling system	Vehicle ownership, average travel distance, and industrial production are the major contributors to PM_2.5_ in PRD.
Yi Yang [12]	2019	China	GDP per capita, industrial added values, urban population density, private car ownership.	Spatial econometric analysis.	GDP per capita, industrial added value and private car ownership are significantly positive to PM_2.5_ concentration, and urban population density

**Table 2 ijerph-17-00074-t002:** The information of all selected variables in this study.

Variable	Full Name	Abbreviation Definition	Unit	Types	Reference
lnPD	Logarithm of the population density	PD: the number of people city divided by area	Pop./km^2^	P (Population)	[9,10,27,28,29,30,31]
lnGDP	Logarithm of gross regional product	GDP: gross regional product of cities	100 million yuan	A (Affluence level)	[29,31,32]
lnGDPP	Logarithm of gross regional product per capita	GDPP: per capita gross regional product	yuan/capita	A (Affluence level)	[12,29,30,31,32,33,34]
lnSIR	Logarithm of the ratio of secondary industry	SIR: the secondary industry divided by total industry output	%	T (Technical level)	[11,29,30,32,33]
lnEC	Logarithm of energy consumption per unit of output	EC: Energy consumption divided by the corresponding output	Tons of standard carbon/10 thousand yuan	T (Technical level)	[13,14],
lnBR	Logarithm of the ratio of urban built-up area	BR: the built-up area divided by city area	%	E (Urban environment)	[35,36]
lnGR	Logarithm of the ratio of green space	GR: the green area divided by city area	%	E (Urban environment)	[28,37]
lnPP	Logarithm of per capita park area	PP: park area divided by population	km^2^/capital	E (Urban environment)	[37]

**Table 3 ijerph-17-00074-t003:** Test results of Global Moran’s I of PM_2.5_ concentration in the CPEZ.

Time	Moran’ I	Standard Error	Z-Score	*p*-Value
2006	0.191 **	0.386	1.655	0.049
2007	0.096 *	0.360	1.461	0.072
2008	0.091 *	0.361	1.379	0.084
2009	0.103 *	0.371	1.522	0.064
2010	0.150 **	0.372	1.728	0.042
2011	0.101 *	0.344	1.405	0.080
2012	0.094 *	0.352	1.580	0.057
2013	0.132 *	0.372	1.607	0.054
2014	0.093 *	0.304	1.379	0.084
2015	0.089 *	0.319	1.491	0.068
2016	0.083 *	0.308	1.483	0.069

Notes: *, ** represent the significance at the 10%, 5%, and 1% level, respectively.

**Table 4 ijerph-17-00074-t004:** The regression results of various effects of SDM.

Models	Model 1	Model 2	Model 3	Model 4
Variables	SDM Time Fixed Effect	SDM Spatial Fixed Effect	SDM Time and Spatial Fixed Effect	SDM Random Effect
lnPD	0.3606 (0.5552) ***	0.1420 (0.1486)	0.2378 (0.1737) *	0.2214 (0.1878) *
lnGDP	0.0770 (0.1186) **	0.0531 (0.0556)	0.2155 (0.1574) *	0.2293 (0.1945)
lnGDPP	0.2068 (0.3184) *	0.0231 (0.0242)	0.0201 (0.0147)	0.0407 (0.0345)
lnSIR	0.0149 (0.0229)	0.1046 (0.1095)	0.0011 (0.0008)	0.0300 (0.0254)
lnEC	0.1350 (0.2079) ***	0.5369 (0.5620) **	0.6741 (0.4925) **	0.5738 (0.4866) **
lnBR	0.0389 (0.0599)	0.0252 (0.0264)	0.1709 (0.1249) *	0.1848 (0.1567)
lnGR	−0.1259 (0.1938) **	−0.1317 (0.1378) **	−0.1519 (0.1110) **	−0.1447 (0.1227) ***
lnPP	−0.1118 (0.1721) ***	−0.0027 (−0.0028)	−0.0144 (0.0105)	−0.0021 (0.0018)
W lnPD	0.10818 (0.16656) ***	0.0426 (0.04458)	0.07134 (0.05211) **	0.06642 (0.05634) *
W lnGDP	0.0231 (0.03558) ***	0.01593 (0.01668)	0.06465 (0.04722) *	0.06879 (0.05835)
W lnGDPP	0.06204 (0.09552) **	0.00693 (0.00726)	0.00603 (0.00441)	0.01221 (0.01035)
W lnSIR	0.00447 (0.00687)	0.03138 (0.03285)	0.00033 (0.00024)	0.009 (0.00762)
W lnEC	0.0405 (0.06237) **	0.16107 (0.1686) ***	0.20223 (0.14775) **	0.17214 (0.14598) **
W lnBR	0.01167 (0.01797)	0.00756 (0.00792)	0.05127 (0.03747)	0.05544 (0.04701)
W lnGR	−0.08777 (0.10814) ***	−0.03951 (0.04134) **	−0.04557 (0.0333) **	−0.04341 (0.03681) ***
W lnPP	−0.09354 (0.11163) **	−0.00081 (0.00084)	−0.00432 (0.00315)	−0.00063 (0.00054)
rho(ρ)	0.154 (0.1831) ***	0.2049 (0.2145) ***	0.2155 (0.1574)	0.146 (0.1390)
R^2^	0.9495	0.8272	0.8601	0.8110
Sig.	0.0061	0.0041	0.0057	0.0063
AdjR^2^	0.6323	0.4416	0.5216	0.4012
observations	88	88	88	88

Notes: Standard errors in parentheses; *, **, *** represent the significance at the 10%, 5%, and 1% level, respectively.

**Table 5 ijerph-17-00074-t005:** The results of the effect decomposition.

Variables	Direct Effect	Indirect Effect	Total Effect
lnPD	0.3606 *** (0.5552)	0.10818 *** (0.1665)	0.46878 *** (0.7217)
lnGDP	0.0770 ** (0.1186)	0.0231 *** (0.0355)	0.1001 ** (0.1541)
lnGDPP	0.2068 * (0.3184)	0.06204 ** (0.0955)	0.26884 * (0.4139)
lnSIR	0.0149 (0.0229)	0.00447 (0.0068)	0.01937 (0.0297)
lnEC	0.1350 *** (0.2079)	0.0405 ** (0.06237)	0.1755 ** (0.27027)
lnBR	0.0389 (0.0599)	0.01167 (0.0179)	0.05057 (0.0778)
lnGR	−0.1259 ** (0.1938)	−0.08777 *** (0.1081)	−0.16367 ** (0.3019)
lnPP	−0.1118 *** (0.1721)	−0.09354 ** (0.1116)	−0.14534 *** (0.2837)

Notes: Standard errors in parentheses; *, **, *** represent the significance at the 10%, 5%, and 1% level, respectively.

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
