# Peer review of "Spatial Econometric Analysis of the Impact of Socioeconomic Factors on PM2.5 Concentration in China’s Inland Cities: A Case Study from Chengdu Plain Economic Zone"

_ijerph, 2019, doi:10.3390/ijerph17010074_

Round 1

Reviewer 1 Report

The manuscript by Yang et al focuses on spatial modlling of PM2.5 concentration in China's Inland cities. I have the following comments for minor revision.

Provide references for the few studies mentioned line 62-64. Table 1 is not cited throughout until the discussion. In the discussion the authors referred to GDP in Table 1, however there is no such information on GDP in table 1. I am of the opinion that Table 1 does not provide any meaningful information and as such it should be deleted or moved to the supplementary materials. The use of table 2 for definition of terms in equation (5) is not conventional. It affects the flow of the manuscript. Line 150: What is SLM? There is no study limitation.

Author Response

Dear reviewers:

We are truly grateful to your and other reviewers’ critical comments and thoughtful suggestions. Based on these comments and suggestions, we have made careful modifications on the original manuscript. All changes made to the text are in red color. We hope the new manuscript will meet your magazine’s standards. Below you will find our point-by-point responses to the reviewers’ comments/ questions:

Point 1: Provide references for the few studies mentioned line 62-64.

Response: Thanks for your advice. We have added some related references which consistent with the reference mentioned in Table 2.

“However, those studies were mainly concentrated in BTH [9], YRD [10], and PRD [11], which are eastern coastal areas of China with high population density and developed economy, while few studies have been conducted in inland cities of China.”

Point 2: Table 1 is not cited throughout until the discussion. In the discussion the authors referred to GDP in Table 1, however there is no such information on GDP in table 1. I am of the opinion that Table 1 does not provide any meaningful information and as such it should be deleted or moved to the supplementary materials.

Response: We apologize for confusing the reviewer with the coding errors in Table 1 and Table S1. Table 1 is a summary of related literature review and we have cited it in line 60-61. “More details about those studies are summarized in Table 1.” Instead of Table 1, in the discussion referred to GDP should have been TableS1, which is the statistical description of socio-economic variables we have selected. Thank reviewer for pointing out the mistake and have corrected accordingly in line 272-273. “It can be seen from Table S1 that the GDP of Chengdu is far higher than that of other cities in CPEZ.”

Point 3: The use of table 2 for definition of terms in equation (5) is not conventional. It affects the flow of the manuscript.

Response: To make equation (5) more conventional, we have deleted the definition of terms in it and the missed definition has added into Table 2.

Point 4: Line 150: What is SLM?

Response: The SLM is the abbreviation of Spatial lag model. Admittedly, we neglected to define SLM to keep the consistency in format, so we have revised it. “For example, common spatial interaction relations include endogenous interactions between dependent variables (Spatial lag model (SLM)), interactions between error terms (spatial error model (SEM)), and based on SLM adding exogenous interactions between independent variables (spatial Durbin model (SDM)), as shown in Figure 3.”

Point 5: There is no study limitation.

Response: Based on the comments of the reviewers and the comparison of relevant studies, the main limitations of this paper are highlighted at the end of the conclusion section. The limitation of this paper is embodied in two aspects. On the one hand, the selection of factors only emphasizes the role of socio-economic impact factors, while ignoring the role of physical environment variables such as wind, temperature and terrain. On the other hand, the number of cities or scope of study area is small, so more samples are needed for further research in the future to improve the credibility of the research results.

“Although the results fill a research gap in the inland cities and put forward a range of meaningful suggestions, there are still some deficiencies that mainly in two aspects. On the one hand, the selection of factors only emphasizes the role of socio-economic impact factors, while ignoring the role of physical environment variables such as wind, temperature and terrain. On the other hand, the number of cites or scope of study area is small, so more samples are needed for further research in the future to improve the credibility of the research results.”

Yours sincerely

Lan Haifeng

Reviewer 2 Report

General comment

The paper regards an econometric study to estimate the impact of some socioeconomic factors on PM2.5 concentration in China's Inland Cities. Spatial and temporal trends are discussed with a possible overview of the implications for mitigation policies. The topic is interesting and suitable for the Journal, however, there are some aspects that are not clear and some things that need to be discussed in more detail (see my specific comments). I believe that the paper could be considered for publication after a major revision.

Specific comments

Line 21. A weak trend is not very quantitative. Please mention if it is statistically significant or not.

Lines 40 and 66. Please remove Etc. If authors want to add something, it is better to do it explicitly.

Lines 52-55- This sentence is not clear and it seems incomplete.

Line 133. It is mentioned PM2.5 emissions but it seems that concentrations are used in the analysis. Please correct or explain.

Line 160. The symbol ρ is not that reported in Equation (6).

Looking at Fig. 4 it seems that there is a decreasing time trend but this does not appear in Fig. 2. Or at least it seems that only 2006 is larger for some areas. I believe that it is necessary a deeper discussion on the time trend if it is statistically significant and for what areas. In addition, the names in the maps are absolutely not readable not even in the screen.

Figure 5. It is not clear what is reported in the x-axis and in the y-axis. Please give some additional explanations.

Table 4. Why the variables are repeated two times? I mean, for example, lnGDP and W*lnGDP. Are two different runs of the model? In the other tables (tables 5 and S1) are reported only once. In addition, it would be better to clearly explain what are the differences among the Model1, Model2, Model3, Model4.

Lines 241-243. Better “However, is relatively low compared….”

Line 243. Too many “:”.

Line 250. “…air exchange relationship…” what it is?

Line 279. Temporospatial is awkward, better temporal and spatial patterns.

Lines 288-299. This sentence is in agreement with Figure 2? See my previous point regarding the trends.

Author Response

Dear reviewers
We are truly grateful to yours and other reviewers’ critical comments and thoughtful suggestions. Based on these comments and suggestions, we have made careful modifications on the original manuscript. All changes made to the text are in red color. We hope the new manuscript will meet the magazine’s standards. Below you will find our point-by-point responses to the reviewers’ comments/ questions:

General comments

Point 1: The paper regards an econometric study to estimate the impact of some socioeconomic factors on PM2.5 concentration in China's Inland Cities. Spatial and temporal trends are discussed with a possible overview of the implications for mitigation policies. The topic is interesting and suitable for the Journal, however, there are some aspects that are not clear and some things that need to be discussed in more detail (see my specific comments). I believe that the paper could be considered for publication after a major revision.

Response: Thanks to the reviewer sincere opinions. We will carefully revise and reply according to reviewer's suggestions, anticipating to be accepted by the journal.

Specific comments

Point 2: Line 21. A weak trend is not very quantitative. Please mention if it is statistically significant or not.

Response: To make the result of trend more quantitative and clearly. We revised the statement.

“Moran’s index indicates that PM2.5 concentration in CPEZ does have spatial aggregation characteristics. In general, the spatial clustering from the fluctuation state to the stable low state decreased by 1% annually on average, from 0.190 (p < 0.05) in 2006 to 0.083(p < 0.1) in 2016.”

Point 3: Lines 40 and 66. Please remove Etc. If authors want to add something, it is better to do it explicitly.

Response: Yes, we have removed etc.

Point 4: Lines 52-55- This sentence is not clear and it seems incomplete.

Response: We changed the sentence to be more clear and complete. “The study of Yun [10] adopted the spatial statistical analysis and the establishment of a geographic detector model to explore the concentrations of PM2.5 in Yangtze River Delta (YRD) from 2005 to 2015, which revealed that population density is the key factor affecting the concentrations of PM2.5.”

Point 5: Line 133-It is mentioned PM2.5 emissions but it seems that concentrations are used in the analysis. Please correct or explain.

Response: Thanks for pointing out this. To make the statement consistent in the context of the article. We have changed the “PM2.5 emissions” to “PM2.5 concentrations”

Point 6: Line 160. The symbol ρ is not that reported in Equation (6).

Response: The ρ is the spatial autocorrelation coefficient of the dependent variable. We have mentioned it below the Equation (6).

Point 7: Looking at Fig. 4 it seems that there is a decreasing time trend but this does not appear in Fig. 2. Or at least it seems that only 2006 is larger for some areas. I believe that it is necessary a deeper discussion on the time trend if it is statistically significant and for what areas. In addition, the names in the maps are absolutely not readable not even in the screen.

Response: Admittedly, the reviews opinion is pretty good. We examined the problem carefully and redraw the Fig.4. The new one is much consistent with the Fig. 2. Additionally, we have adopted reviewer suggestion and demonstrate cluster and outlier analysis of PM2.5 concentrations from 2006-2016 in Fig 5. By the figure we can clearly see the where is the statistically significant cluster or outlier, how is it change with time. The related discussion as following.

 “In order to further explore the changes of spatial aggregation of PM2.5 concentration with time, a cluster and outlier analysis of PM2.5 concentrations from 2006-2016 is adopted, as shown in Figure 5. By the figure we can clearly see that the cluster areas with statistical significance (p < 0.05) are mainly Chengdu and cities in the east of Chengdu such as Deyang, Ziyang and Suining. While, the outlier areas with statistical significance (p < 0.05) are mainly located at the edge of CPEZ, where is relatively far from Chengdu. From the perspective of temporal variation trend, the spatial distribution of cluster and outlier before 2012 showed an obvious change state. Almost every year is different. While in the recent four years, the spatial distribution of cluster and outlier showed a stable state.”

Point 8: Figure 6. It is not clear what is reported in the x-axis and in the y-axis. Please give some additional explanations.

Response: We have added some introduction about Moran scatter plot and the additional explanations as following.

The Moran scatter plot [39] is a useful visual tool for exploratory analysis, because it enables you to assess how similar an observed value is to its neighboring observations. Its horizontal axis is based on the values of the observations and is also known as the response axis. The vertical Y axis is based on the weighted average or spatial lag of the corresponding observation on the horizontal X axis.”

Point 9: Table 4. Why the variables are repeated two times? I mean, for example, lnGDP and W*lnGDP. Are two different runs of the model? In the other tables (tables 5 and S1) are reported only once.

Response: In this research we adopted the SDM (spatial During Model), which can reflect the

direct effect and indirect effect (spatial spillover effect) of variables as mentioned in the part of 2.4. Spatial econometric model. For example, in table 4 the coefficient of lnGDP means direct effect of GDP to the concentration of PM2.5, while W*lnGDP represent the indirect effect of GDP to the concentration of PM2.5. In table 5, the variables actually are also reported twice, because it contains direct effect and indirect effect. Additionally, those all coefficients of variables no matter direct or indirect effect can be calculated based on the data of TableS1. So, it just should be reported once.

Point 10: In addition, it would be better to clearly explain what are the differences among the Model1, Model2, Model3, Model4.

ResponseIn order to make those models to be better understand, we have briefly introduced those models.

“The purpose of the model (SDM) is to identify the impact of social-economic variables on PM2.5 concentration which contain the dimensions of space and time, but we cannot determine directly whether the difference in PM2.5 concentration caused by time and space is random (random effect) or presenting a certain regularity (fixed effect), so we adopt four models for comparison and a brief introduction of those models as following.”

(1) SDM time fixed effect: for different spatial individuals, differences caused by time are consistent.

(2) SDM spatial fixed effect: among cross-sectional data of different time series, differences caused by spatial characteristics are consistent.

(3) SDM time and spatial fixed effect: among cross-sectional data of different time series, differences caused by space are consistent, and among different spatial individuals, differences caused by time are consistent.

(4) SDM random effect: the differences caused by space and time are random.

Point 11: Lines 241-243. Better “However, is relatively low compared….”

Response: Yes, we change this sentence to “However, Moran’s I of PM2.5 concentration in CPEZ is relatively lower compared with the three major economic growth areas (BTH, YRD, and PRD), which is about 0.4 to 0.9.”.

Point 12: Line 243. Too many “:”.

Response: We have deleted some superfluous “:”, and reorganized the sentence. “For Economic reason, it can be seen from Table S1 that the GDP of Chengdu is far higher than that of other cities in CPEZ.”

Point 13: Line 250. “…air exchange relationship…” what it is?

Response: Actually, it means air flow. We change it to “This kind of terrain makes it difficult for the central city group to form air flow with its peripheral cities.”.

Point 14: Line 279. Temporospatial is awkward, better temporal and spatial patterns.

Response: We have replaced “temporospatial” with “temporal and spatial patterns”.

Point 15: Lines 288-289. This sentence is in agreement with Figure 2? See my previous point regarding the trends.

Response: We examined the problem again and we have changed it to a more accurate statement. “In addition, from 2006 to 2016, the spatial aggregation characteristics showed that the initial high and low fluctuation states gradually changed to stable low fluctuation states.”

Yours sincerely

Lan Haifeng

Reviewer 3 Report

The authors propose performing a spatial econometric analysis of socioeconomic factors in the concentration of PM2.5 in islands of China. They consider answering three questions. The study is carried out in 8 cities and describes some general characteristics. However, I consider that more details should be given on some apsects such as the number of inhabitants of each city of the cities, main anthopogenic activities, if possible an estimated number of industries, vehicles etc., in the same way it would be interesting that some parameters could be included. meteorological conditions (temperature, relative humidity, wind speed and direction), because in line 87 (section 2.1), the authors give a reference on the low diffusion and dilution of air pollutants; however, it would be constructive for the reader to briefly explain the impact of these variables on the concentration of PM2.5.

On the other hand, in table 2, the authors mention the selected variables; and again it is suggested that they annex the quantities used for each of them in the different cities studied. Finally, it would be interesting for the authors to compare their results with others observed, in other cities of the Chinese Republic

Author Response

Dear reviewers
We are truly grateful to yours and other reviewers’ critical comments and thoughtful suggestions. Based on these comments and suggestions, we have made careful modifications on the original manuscript. All changes made to the text are in red color. We hope the new manuscript will meet the magazine’s standards. Below you will find our point-by-point responses to the reviewers’ comments/ questions:

Point 1: The authors propose performing a spatial econometric analysis of socioeconomic factors in the concentration of PM2.5 in islands of China. They consider answering three questions. The study is carried out in 8 cities and describes some general characteristics. However, I consider that more details should be given on some aspects such as the number of inhabitants of each city of the cities, main anthropogenic activities, if possible an estimated number of industries, vehicles etc., in the same way it would be interesting that some parameters could be included. meteorological conditions (temperature, relative humidity, wind speed and direction), because in line 87 (section 2.1), the authors give a reference on the low diffusion and dilution of air pollutants; however, it would be constructive for the reader to briefly explain the impact of these variables on the concentration of PM2.5.

Response: The suggestions put forward by reviewer are exactly what we wanted to study at the beginning, but due to the limitations of data availability and method feasibility, we finally decided to focus on the impact of socio-economic indicators on PM2.5 concentration. Specifically, the choice from a socio-economic perspective is based on two considerations. On the one hand, although many studies reveal that the physical environmental conditions in cities have a great influence on the concentration of PM2.5, more studies show that the concentration of PM2.5 is mainly closely related to human social and economic activities. On the other hand, it is because of access and compatibility issues with data. As we know that the majority the physical environment data (such as temperature, humidity, radiation, wind speed and direction) are obtained from ground weather stations, which belong to the point feature of spatial data. A city often has many meteorological observation points. But the social and economic data are based on the city's administrative statistical boundary, which belong to the polygons feature data. In general, the polygonal feature data cannot correspond to the point feature data, especially at the regional scale. However, we didn’t find good way to solve this problem, thus this paper just explores the influence of socio-economic indicators on PM2.5 concentration. And those limitations would be addressed at the end of article.

“Although the results fill a research gap in the inland cities and put forward a range of meaningful suggestions, there are still some deficiencies that mainly in two aspects. On the one hand, the selection of factors only emphasizes the role of socio-economic impact factors, while ignoring the role of physical environment variables such as wind, temperature and terrain. On the other hand, the number of cities or scope of study area is small, so more samples are needed for further research in the future to improve the credibility of the research results.”

In addition, as mentioned by the reviewer, it would be beneficial to briefly explain the effect of physical environment variables on PM2.5 concentration. In order to make up for the shortcomings of this study, we added some references, aiming at giving readers an overview of how these factors affect PM2.5 concentration.

“Stable weather is easily formed by this geographic environment, which means it is not conducive to the diffusion and dilution of pollutants and aggravates air pollution. Specifically, because CPEZ is located in the basin topography, the atmospheric environment capacity is very limited and where prolonged breezes or calm winds in the area can inhibit advection transport of pollutants and hinder their diffusion, which would multiply the amount of pollutants near the ground [21] Besides, the phenomenon of temperature inversion in urban is serious, that is the upper air temperature is higher than the lower air temperature. Once the formation of this inversion, the air cannot convection up and down, which is difficult to diffuse pollutants[21,22]. Those are why it is one of the 4 regions with the worst smog in China [23].”

Furthermore, the reviewer suggests that we should use variables such as number of inhabitants, number of industries, vehicles, etc.. However, in fact, the variable selected from STIRPAT model in this paper can actually reflect the above indexes to a certain extent. For example, the population density, which is selected by this paper, can reflect more than the number of inhabitants in the intensity of human activities. In the same way, the ratio of the secondary industry can better reflect the development level of urban industries than the number of industries. And the number of vehicles is also highly correlated with the size of the population and the level of the economy. Although it is undeniable that there may be slight differences between the results of different indicators, we believe that the finding of this paper is of guiding and realistic significance to prevent and control of PM2.5 pollution in CPZE.

Point 2: On the other hand, in table 2, the authors mention the selected variables; and again, it is suggested that they annex the quantities used for each of them in the different cities studied. Finally, it would be interesting for the authors to compare their results with others observed, in other cities of the Chinese Republic.

Response: This is a very good suggestion. As the suggestions, we make the comparison between CPEZ and BTH, YRD and PRD. The detail is shown as follows.

“Finally, in order to make the comparison between CPEZ and BTH [9], YRD [10], and PRD [11] more intuitive, we sorted out the regression coefficients obtained from relevant studies according to population, affluence level, technical level and urban environment, and normalized the results, as shown in Figure 8. It is clearly that although there are differences in the weight of influencing factors in each region, the population factor is the most important influencing factor. Additionally, in CPEZ the weight of the affluence level factor is higher than that of other regions, while the weight of the technology level factor is lower than that of other regions. This means the economy of CPEZ is less value-added and energy-efficient than elsewhere and the development of green economy and high value-added industries needs to be strengthened. The influence factors of urban environment lie in the middle level between these regions, indicating that the urban ecological construction performs well but still has room for improvement.”

Yours sincerely

Lan Haifeng

Round 2

Reviewer 3 Report

The authors took into account the recommendations